Ensemble techniques for detecting profile cloning attacks in online social networks

http://orcid.org/0000-0001-8316-8439 Mohiuddin Irfan irfanm@ksu.edu.sa
http://orcid.org/0000-0002-8253-9709 Almogren Ahmad ahalmogren@ksu.edu.sa
Chair of Cyber Security, Department of Computer Science, College of Computer and Information Sciences, King Saud University , Riyadh , Saudi Arabia
Kong Xiangjie
Electronic publication date: 2025 Sep 4
Publication date: 2025
Volume: 11
Electronic Location ID: e3182
Received 2025 Feb 5; Accepted 2025 Aug 11
Copyright: © 2025 Mohiuddin and Almogren
Copyright year: 2025
Copyright holder: Mohiuddin and Almogren
License: This is an open access article distributed under the terms of the Creative Commons Attribution License, which permits unrestricted use, distribution, reproduction and adaptation in any medium and for any purpose provided that it is properly attributed. For attribution, the original author(s), title, publication source (PeerJ Computer Science) and either DOI or URL of the article must be cited.
License URL: https://creativecommons.org/licenses/by/4.0/

Keywords: Profile cloning detection, Ensemble learning, Online social networks, Stylometric analysis, Perplexity scoring, Anomaly detection, Semantic embeddings, Meta-ensemble classifier, Attention-based aggregation

Funding: Deanship of Scientific Research, King Saud University Vice Deanship of Scientific Research Chairs, Chair of Cybersecurity This work was supported by the Deanship of Scientific Research, King Saud University, through the Vice Deanship of Scientific Research Chairs, Chair of Cybersecurity. The funders had no role in study design, data collection and analysis, decision to publish, or preparation of the manuscript.

==============================
Detecting cloned and impersonated profiles on online social networks (OSNs) has become an increasingly critical challenge, particularly with the proliferation of AI-generated content that closely emulates human communication patterns. Traditional identity deception detection methods are proving inadequate against adversaries who exploit large language models (LLMs) to craft syntactically accurate and semantically plausible fake profiles. This article focuses on the detection of profile cloning on LinkedIn by introducing a multi-stage, content-based detection framework that classifies profiles into four distinct categories: legitimate profiles, human-cloned profiles, LLM-generated legitimate profiles, and LLM-generated cloned profiles. The proposed framework integrates multiple analytical layers, including semantic representation learning through attention-based section embedding aggregation, linguistic style modeling using stylometric-perplexity features, anomaly scoring via cluster-based outlier detection, and ensemble classification through out-of-fold stacking. Experiments conducted on a publicly available dataset comprising 3,600 profiles demonstrate that the proposed meta-ensemble model consistently outperforms competitive baselines, achieving macro-averaged accuracy, precision, recall, and F1-scores above 96%. These results highlight the effectiveness of leveraging a combination of semantic, stylistic, and probabilistic signals to detect both human-crafted and artificial intelligence (AI)-generated impersonation attempts. Overall, this work presents a robust and scalable content-driven methodology for identity deception detection in contemporary OSNs.

Introduction

Online social networks (OSNs) play a pivotal role in facilitating global communication and connectivity. However, the proliferation of cloned profiles poses a significant threat to the security and trustworthiness of these platforms. Such malicious profiles that are created manually, by bots, or increasingly through advanced large language models (LLMs) are frequently leveraged for harmful activities including identity theft, misinformation dissemination, and social engineering attacks (Habib et al., 2024). Traditional detection systems often fall short in identifying these profiles, particularly when adversaries exploit LLMs to generate highly realistic and linguistically coherent content.

Previous research has investigated a range of techniques for detecting cloned profiles on OSNs. Early efforts predominantly employed rule-based systems and heuristic-driven methods (Kanagavalli & Priya, 2022). These were later augmented by machine learning approaches that utilized features derived from user behavior, network topology, and textual content to improve classification accuracy (Bhattacharya et al., 2022). However, the advent of LLM-generated profiles has introduced new complexities, as such profiles often exhibit sophisticated, coherent, and semantically rich narratives that convincingly mimic legitimate users—making them increasingly difficult to detect using traditional methods (Kenny et al., 2024).

Recent advancements in natural language processing have underscored the effectiveness of text embeddings in distinguishing between genuine and cloned profiles. Semantic embedding techniques—such as Global Vectors for Word Representation (GloVe), Flair, Bidirectional Encoder Representations from Transformers (BERT), and Robustly Optimized BERT Pretraining Approach (RoBERTa)—have shown considerable promise in this domain, yielding notable improvements in detection accuracy (Shah, Varshney & Mehrotra, 2024; Terumalasetti, 2024). Nevertheless, many existing models treat the entire profile as a single, flattened textual sequence, disregarding the inherent structural heterogeneity of profile sections like “Experience,” “Education,” and “About.” This simplification leads to the loss of valuable contextual and section-specific signals that could otherwise enhance detection performance.

In this article, we address these limitations by proposing a comprehensive, multi-stage classification framework for detecting four distinct types of LinkedIn-style profiles: (1) Legitimate LinkedIn Profiles (LLP), (2) Human-Cloned Profiles (HCP), (3) ChatGPT-Generated Legitimate Profiles (CLP), and (4) ChatGPT-Generated Cloned Profiles (CCP). At the core of our approach is a novel attention-based aggregation mechanism that dynamically weights section-wise semantic embeddings to construct contextually enriched and discriminative profile representations. Unlike simple averaging, this method learns to emphasize the most informative sections, thereby improving classification accuracy and robustness.

We further introduce a stylometric and pseudo-perplexity-based feature modality-referred to as the stylo-perplexity method-that captures linguistic irregularities and coherence deviations often indicative of synthetically generated content. Additionally, we integrate an unsupervised anomaly detection module based on cluster distance scoring, which assigns outlier scores in a feature-agnostic manner to flag profiles exhibiting atypical characteristics.

These features are combined into a unified representation and fed into a meta-classifier trained using out-of-fold (OOF) stacking. This ensemble approach aggregates predictions from a diverse array of base models-including neural networks, tree-based algorithms, and support vector machines—enhancing both robustness and generalization across various profile types.

To support reproducibility and fault tolerance, we employ checkpointing mechanisms during model training. The proposed framework is evaluated on a dataset comprising 3,600 LinkedIn-style profiles-both real and synthetically generated-using only publicly available data to maintain ethical compliance and ensure practical applicability.

Experimental results indicate that our system achieves a macro-averaged F1-score of 96.08% and an accuracy of 96.11%, outperforming individual classifiers and existing baselines. Notably, the model demonstrates high recall on the most challenging classes—HCP and CCP—despite their limited representation in the dataset. These findings underscore the effectiveness of integrating semantic, stylistic, and probabilistic features for robust identity deception detection.

In summary, the key contributions of this work are as follows: (i) a novel attention-based mechanism for aggregating section-wise semantic embeddings, allowing the model to emphasize the most informative components of user profiles; (ii) a new feature fusion strategy that integrates stylometric indicators with probabilistic coherence measures to effectively detect LLM-generated content; (iii) an unsupervised anomaly detection module based on cluster-distance scoring, capable of identifying profiles that deviate from typical distribution patterns; and (iv) a meta-ensemble classification architecture trained via OOF stacking to enhance generalizability and robustness. Collectively, these components form a scalable and high-performing framework for detecting and mitigating modern profile cloning threats in online social networks.

The remainder of the article is structured as follows. ‘Literature Review’ presents a review of the existing literature on cloned profile detection and related techniques. ‘Data Collection’ details the data collection process and describes the characteristics of the dataset. ‘Methdology’ outlines the proposed detection framework, including feature engineering, embedding strategies, and classification architecture. ‘Experimental Overview’ provides an overview of the experimental setup, while ‘Results and Discussion’ reports and discusses the empirical findings. Finally, ‘Conclusion and Future Work’ concludes the article and suggests directions for future research.

Literature review

The challenge of detecting cloned profiles on OSNs has garnered significant research attention in recent years, with methodologies evolving from traditional machine learning techniques to more advanced, artificial intelligence-driven solutions.

Initial efforts in clone profile detection predominantly concentrated on feature extraction and optimization. A variety of studies have proposed detection mechanisms targeting malicious activities in OSNs and related domains. For instance, Siddique et al. (2021) developed a machine learning-based approach for spam email detection, underscoring the importance of effective feature selection in achieving robust classification outcomes. In the context of the Internet of Medical Things (IoMT), Almogren et al. (2020) introduced a fuzzy-based trust management framework to mitigate Sybil attacks, emphasizing the role of trust modeling in countering impersonation threats. Similarly, Masood et al. (2019) employed graph-based techniques to detect spammers and fake users in social networks by analyzing behavioral patterns. To enhance OSN security, Al-Qurishi et al. (2018) proposed a lightweight key agreement protocol to defend against Sybil attacks, while Almasoud et al. (2018) designed advanced privacy-preserving protocols to protect user data.

In the domain of user profiling, Joshi et al. (2021) leveraged machine learning models incorporating social network attributes to identify suspicious accounts, and (Rawat et al., 2021) applied social network analysis methods to examine deceptive user behavior. Complementing these technical advancements, recent surveys provide broader overviews: Alharbi et al. (2021) systematically categorized identity deception tactics and existing detection gaps, whereas (Mostafa et al., 2024) offered a comprehensive review of deep learning frameworks for fake news detection on social media platforms.

A prominent trend in recent years has been the adoption of advanced semantic embedding techniques to represent profile text and posts within detection models. Traditional methods based on bag-of-words and keyword matching are increasingly being replaced by deep neural language models that capture richer contextual and semantic nuances. For example, Goyal & Mahmoud (2024) detect fake Instagram profiles by encoding user biographies using DistilBERT embeddings, which are then input into a Random Forest classifier-achieving an accuracy of approximately 84%. Similarly, Hayawi, Al-Hashimi & Alawjar (2025) propose the “DeeProBot” framework, which embeds Twitter profile descriptions using GloVe vectors and integrates them with numerical metadata in a deep long short-term memory (LSTM)-based architecture. Wanda & Jie (2020) present DeepProfile, a convolutional neural network (CNN)-based classifier designed to extract salient textual patterns from profile content. Collectively, these methods exemplify a broader shift toward representation learning, enabling models to achieve deeper semantic and contextual understanding that surpasses the performance of conventional keyword-based techniques.

In parallel with semantic embeddings, researchers have increasingly investigated the use of linguistic fingerprints embedded within profile content. Stylometric analysis conceptualizes writing style as a distinctive and identifiable signature-even short text snippets can reveal critical clues about whether the language is human-authored or machine-generated. For instance, Kaviya, Sudharsana & Bala Chibi Hariesh (2024) integrated linguistic cues such as atypical phrasing and irregular sentence structure into their detection pipeline, leading to significant improvements in classification performance. Stylometric approaches also analyze features including lexical richness, function word frequency, and textual readability to differentiate between authentic users and bots or impersonators.

Studies such as He et al. (2021) emphasize that stylistic attributes-such as tone, sentiment, and consistency of writing-offer strong indicators for detecting fake profiles. These linguistic analyses serve as valuable complements to semantic embeddings, enhancing both interpretability and robustness in identity deception detection frameworks.

Several studies have explored behavioral modeling and hybrid techniques to enhance the detection of fake or cloned accounts. Wang, Zhu & Yang (2021) proposed composite behavioral models that utilize user interaction patterns to identify cases of identity theft. Bharti & Pandey (2021) introduced a hybrid detection method that combines logistic regression with particle swarm optimization, resulting in improved classification accuracy. Similarly, Kanagavalli & Priya (2022) developed a deep learning framework that integrates both behavioral and contextual features, while (Chakraborty et al., 2022) employed customized feature engineering strategies tailored to specific attack vectors. Optimization-based enhancements have also been investigated, as demonstrated by Wanda (2022), who improved CNN-based classifiers through the adoption of novel activation functions.

Ensemble and hybrid architectures have garnered increasing attention for their ability to integrate diverse detection models and heterogeneous feature types. Kaviya, Sudharsana & Bala Chibi Hariesh (2024) evaluated ensemble classifiers, including Random Forests and voting-based schemes, and reported superior accuracy compared to individual base learners. The DeeProBot architecture exemplifies the effectiveness of hybrid integration by combining LSTM-based textual analysis with numerical metadata, thereby capturing both semantic and structural characteristics of user profiles. Similarly, He et al. (2021) proposed a multi-factor attention model that incorporates bidirectional LSTM (Bi-LSTM) streams to process both static and dynamic textual features. These strategies address the limitations of single-model detectors and significantly improve the detection of sophisticated fake profiles that closely imitate legitimate user behavior.

Recent studies have proposed innovative approaches that broaden the scope of fake profile detection by incorporating deep learning and multimodal techniques. Mughaid et al. (2023) integrated facial recognition with machine learning algorithms to strengthen identity verification mechanisms in social networks. Aditya & Mohanty (2023) examined the unique challenges associated with detecting fraudulent behavior across heterogeneous social platforms. In a related effort, Nevado-Catalán et al. (2023) conducted an in-depth analysis of fake engagement services, offering critical insights into the operational strategies employed by fraudulent accounts.

Honeypot-based detection systems enhanced by deep learning were introduced by El Mendili et al. (2024), demonstrating strong potential for identifying malicious entities. Transformer-based embeddings were effectively employed by Ramdas & Agnes (2024) and Shah, Varshney & Mehrotra (2024) to extract semantically rich features that improve classification performance. In addition, Terumalasetti (2024) applied optimization strategies to improve the training efficacy of deep neural networks in the context of social network security.

Comprehensive surveys by Habib et al. (2024) and Kenny et al. (2024) have identified persistent limitations in hybrid detection models and underscored the influence of cognitive biases on human judgment-highlighting the need for improved user education and the development of explainable AI-based detection frameworks.

The emergence of LLMs such as GPT-3, GPT-4, and ChatGPT has facilitated the generation of synthetic profiles that closely resemble authentic user accounts, making them increasingly challenging to detect. Ferrara (2023) and Cresci et al. (2023) underscore the threats posed by ChatGPT-enabled bots and advocate for detection strategies that incorporate generative AI as part of the defensive toolkit. In response to these developments, researchers have begun exploring techniques such as stylometric analysis, perplexity-based scoring, and adversarial interrogation to identify AI-generated content. This rapidly evolving threat landscape calls for the development of adaptive detection frameworks that are explicitly attuned to the linguistic and structural patterns characteristic of LLM-generated profiles and are capable of evolving in tandem with advances in generative model capabilities.

Despite significant progress in fake profile detection, several critical research gaps persist. Many existing detection models are developed and validated using data from a single platform—most commonly Twitter-and are typically restricted to a single language, often English. Consequently, these models often exhibit diminished performance when applied to alternative platforms such as LinkedIn or when analyzing content in other languages.

To enhance generalizability, recent research has emphasized the extraction of domain-independent textual features and the application of transfer learning techniques to adapt models across varying social media environments. A promising direction involves training multilingual, transformer-based models on diverse social media corpora to learn representations that remain effective across different platforms, linguistic contexts, and user behaviors.

Early detection remains another critical challenge, wherein systems must accurately identify malicious profiles based on limited input data-prompting the exploration of few-shot and one-shot learning paradigms. These approaches aim to detect threats such as impersonation using minimal information, for example, identifying a fake profile from a single description or a small set of messages. By leveraging the rich prior knowledge embedded in pretrained language models, such methods hold promise for enabling rapid and accurate detection in scenarios where data availability is constrained.

Robustness to adversarial manipulation is another essential requirement, as malicious actors may employ tactics such as paraphrasing, obfuscation, or the injection of textual noise to bypass detection mechanisms. To address this challenge, recent research has explored adversarial training techniques, wherein models are exposed to synthetically modified text that simulates common evasion strategies. Concurrently, efforts are underway to improve model resilience by leveraging deeper semantic representations that remain invariant to surface-level alterations. For instance, even if a cloned profile rephrases or rearranges sentence structures to evade detection, a context-aware embedding model can still uncover semantic inconsistencies between the profile’s textual content and the identity it purports to represent.

The issue of model explainability remains a persistent challenge, with growing demand for interpretable detection systems that promote transparency and trust among end users, researchers, and system administrators. As detection models become more complex, the ability to provide meaningful explanations for classification decisions becomes increasingly important for fostering accountability and facilitating human oversight.

Moreover, there is a rising recognition of the benefits of multimodal fusion, wherein textual data is combined with complementary modalities-such as visual cues for example profile images and structural information for example social network graphs to construct more comprehensive and resilient detection frameworks.

In summary, the existing literature reflects a clear progression from rule-based heuristics to more advanced methodologies grounded in semantic representation, stylistic analysis, and ensemble-based learning. In light of the increasing prevalence of LLM-generated content, the field is progressively adopting integrative strategies that combine deep representation learning, stylometric profiling, and multimodal feature fusion. These approaches are pivotal in addressing the growing sophistication and complexity of online impersonation threats across diverse social platforms.

Data collection

The dataset utilized in this study is a publicly available collection of LinkedIn profiles, originally compiled and annotated by the authors in Ayoobi, Shahriar & Mukherjee (2023). Each profile comprises multiple textual sections-such as professional background, education, achievements, and skills-that reflect the structured format typical of real LinkedIn profiles. The dataset is annotated with target labels indicating whether a profile is legitimate, manually cloned, or generated using a LLM. This well-annotated and diverse dataset offers a robust foundation for evaluating the proposed ensemble-based detection framework across a range of profile types.

The dataset consists of 3,600 professional LinkedIn profiles, categorized into four distinct classes: 1,800 legitimate profiles, 600 manually created cloned profiles, 600 LLM-generated legitimate profiles (i.e., authentic users who utilized a large language model to generate their content), and 600 LLM-generated cloned profiles. Each profile contains rich textual information distributed across multiple sections, including work experience, education, skills, and achievements.

The feature selection strategy employed in this article is guided by two primary considerations. First, the objective is to facilitate the early detection of cloned accounts immediately upon their creation-prior to the establishment of any social connections-thereby preventing impostors from gaining access to sensitive or private information associated with legitimate users. Second, due to privacy constraints, the analysis is restricted to publicly available profile information, excluding any data that is accessible exclusively to a user’s connections. These constraints ensure that the proposed detection framework is both privacy-preserving and applicable in real-world deployment scenarios where access to private user data is not guaranteed.

As a result, the dataset includes only publicly visible profile attributes that are accessible to any LinkedIn user, thereby ensuring full compliance with privacy standards. These attributes encompass general metadata such as workplace, location, number of connections and followers, and the presence or absence of a profile picture. In addition, the dataset captures the textual content of various profile sections, including About, Experiences, Education, Licenses, Volunteer Work, Skills, Recommendations, Projects, Publications, Courses, Honors and Awards, Scores, Languages, Organizations, Interests, and Activities. This comprehensive selection of publicly accessible content supports the robust detection of fake and cloned accounts while maintaining strict adherence to user privacy constraints.

Ayoobi, Shahriar & Mukherjee (2023) identified fake profiles using a multi-faceted strategy. They began by searching LinkedIn posts containing hashtags such as #fake_accounts, #fake_profiles, #scammers, #spammers, and #bot, which led them to user-reported complaints about suspicious accounts. In addition, they collected fake profiles that had been directly reported by LinkedIn users who experienced unusually high volumes of connection requests from such accounts. To ensure the reliability of the dataset, all identified profiles were manually verified by the authors.

The dataset also includes fake profiles generated by organizations attempting to artificially inflate employee counts through the creation of fraudulent accounts. These were uncovered through targeted searches involving specific job titles and company names, using the Bing search engine to identify anomalous patterns in profile listings.

Ayoobi, Shahriar & Mukherjee (2023) sought to anticipate emerging challenges in fake profile detection by synthetically generating 1,200 profiles using ChatGPT. Their approach was based on the hypothesis that users will increasingly rely on LLMs to complete profile sections that were traditionally filled out manually. To enhance the realism of the generated content, the authors began by sampling structural statistics-such as the number of entries within a given section-from both legitimate and fake profiles. These statistics informed the design of prompts used to guide ChatGPT in producing realistic and contextually appropriate text.

For instance, to populate the Experiences section with three entries, a prompt was constructed instructing ChatGPT to generate three professional experiences, each comprising a role, job title, company name, start and end dates, duration, workplace location (city, state, country), and a concise description of responsibilities. This method facilitated the generation of semantically rich and structurally coherent profile content that closely mirrors authentic user behavior.

We concur with the anticipation of Ayoobi, Shahriar & Mukherjee (2023) that, in the near future, individuals are likely to increasingly rely on LLMs to generate profile content-particularly in sections that were traditionally completed manually. However, a key distinction in our research on cloned profile detection lies in the adopted preprocessing strategy, embedding methodology, and classification framework. Our approach specifically emphasizes the exploitation of stylistic and linguistic variations between human-authored and LLM-generated text to differentiate among legitimate profiles, manually cloned profiles, and LLM-generated counterparts.

In contrast, the preprocessing techniques employed by Ayoobi, Shahriar & Mukherjee (2023)—including lowercasing capitalized words, attributive alterations, and stop-word removal—may have inadvertently hindered model performance. These steps risk eliminating semantically and stylistically informative cues, such as capitalization patterns and function word usage, which are critical for identifying subtle inconsistencies in synthetically generated content. This highlights the importance of task-aware preprocessing, especially in applications where linguistic fidelity and stylistic markers play a central role in effective detection.

The proposed framework is designed to transform multi-section user profiles into a unified semantic representation through an attention-based aggregation mechanism. This approach enables the model to selectively emphasize the most informative profile sections, rather than treating all segments with equal weight. To further strengthen detection capabilities, we incorporate stylo-perplexity features, which jointly capture linguistic style and probabilistic coherence-facilitating the identification of subtle writing patterns that may suggest synthetic or manipulated content.

In addition, the framework integrates anomaly scoring based on cluster distance metrics to detect outlier profiles that deviate from normative patterns. This component enhances the system’s ability to flag suspicious accounts, even in the absence of labeled data. Finally, our ensemble classification strategy employs OOF stacking to combine predictions from a diverse set of base classifiers, thereby improving robustness, generalization, and accuracy across multiple profile categories.

Methodology

In this section, we present the proposed methodology for detecting legitimate profiles, manually cloned profiles, and LLM-generated LinkedIn profiles. Our approach comprises five key components: (i) context-preserving semantic embeddings for individual profile sections, (ii) a learnable attention-based aggregation mechanism to generate unified representations, (iii) stylometric and perplexity-based linguistic feature extraction to capture stylistic and coherence-related patterns, (iv) unsupervised anomaly detection in the learned feature space using cluster-based distance metrics, and (v) a robust ensemble classification framework employing OOF stacking. Each module is designed to extract and fuse complementary signals indicative of identity deception or synthetic content, thereby enhancing the system’s accuracy, robustness, and generalizability.

Problem formulation

Let D={(Xi,yi)}i=1N denote the dataset, where Xi represents a structured social network profile consisting of M textual sections, and yi∈{0,1,2,3} is the class label corresponding to one of four profile types: LLP, HCP, CLP, and CCP. The objective is to learn a multi-class classifier f:X→{0,1,2,3} that generalizes to unseen profiles.

Section-wise semantic embedding and aggregation

Each profile Xi={Xi(1),Xi(2),…,Xi(M)} comprises M sections, such as “About,” “Experience,” and “Skills.” We denote by ϕ(⋅):X→Rd a pre-trained sentence embedding function (e.g., Sentence-BERT), which maps each section to a d-dimensional semantic vector:

ei,j=ϕ(Xi(j)),∀j∈{1,…,M}.

We stack these embeddings into a matrix Ei∈RM×d, where each row represents the embedding of a specific section.

To obtain a single representation for the entire profile, we introduce a learnable attention mechanism that aggregates section embeddings using attention weights:

αi,j=exp⁡(ei,j⊤q)∑j′=1Mexp(ei,j′⊤q),

where q∈Rd is a trainable query vector shared across profiles. The aggregated profile embedding is then computed as:

hi=∑j=1Mαi,jei,j∈Rd.

This formulation allows the model to dynamically emphasize the most informative sections, such as those where LLM-generated content tends to diverge from human writing.

This learned attention mechanism emphasizes discriminative sections (e.g., unusual “About” descriptions in clone profiles or inconsistent “Skills” in LLM-generated profiles). The process of aggregating section embeddings into a single profile embedding using attention weights is detailed in Algorithm 1. This algorithm highlights the use of a learnable query vector to assign weights to different sections, emphasizing those that are more informative for authenticity assessment.

Algorithm 1 Attention-based aggregation for profile embeddings.

Require: Stacked section embeddings Ei=[ei,1,ei,2,…,ei,M]∈RM×d for profile i, query vector q∈Rd	
Ensure: Aggregated profile embedding hi∈Rd	
 1: Step 1: Compute Attention Scores	
 2: for j=1 to M do	
 3:    Compute unnormalized attention score:	
              si,j=ei,j⋅q⊤	
 4: end for	
 5: Compute normalized attention weights:	
         αi,j=exp⁡(si,j)∑j′=1Mexp(si,j′),∀j∈{1,…,M}	
 6: Step 2: Aggregate Section Embeddings	
 7: Compute the aggregated profile embedding:	
         hi=∑j=1Mαi,jei,j	
       return hi	

Stylometric feature construction

In parallel, we extract a stylometric feature vector si∈Rp from the full concatenated profile text Xifull=⊕j=1MXi(j), where ⊕ denotes textual concatenation. These features capture various linguistic signals across multiple dimensions. Lexical features include average word length and vocabulary richness, which help characterize the diversity and complexity of word usage. Syntactic features such as sentence length variance and part-of-speech (POS) tag entropy provide insights into the structural patterns of the text. Orthographic features—including punctuation frequency and the ratio of uppercase letters—highlight surface-level stylistic elements. Finally, readability metrics like the Flesch Reading Ease and Gunning Fog Index quantify the linguistic accessibility of the profile content, offering further cues about whether the text was authored naturally or generated artificially. Formally,

si=[s1(Xifull),…,sp(Xifull)].

Perplexity-based fluency scoring

We introduce a probabilistic linguistic coherence score based on masked language modeling (MLM). Let Ti=(w1,…,wL) denote the tokenized profile text truncated to length L. For a subset of masked positions M⊂{1,…,L}, we compute the average negative log-likelihood of predicting masked tokens:

pi=1|M|∑t∈M−log⁡P(wt∣Ti(−t)),

where Ti(−t) denotes the input with the t-th token masked. This scalar value pi∈R serves as a fluency or perplexity-based anomaly signal: LLM-generated profiles often yield smoother distributions with low variability in pi.

Anomaly scoring via cluster distances

To incorporate distributional deviation, we perform unsupervised clustering over the combined features (hi,si,pi) using K-means. Let {c1,…,cK} denote the cluster centroids. For each profile, we define its anomaly score as the Euclidean distance to the nearest cluster center:

di=mink∈{1,…,K}||fi−ck||2,

where fi=[hi;si;pi]∈Rd+p+1. Profiles with higher di values are flagged as potential outliers.

Final feature vector and classifier input

The final feature vector for profile i is defined as:

xi=[hi;si;pi;di]∈Rd+p+2.

These features collectively encode semantic, stylistic, probabilistic, and anomaly-based perspectives, allowing the classifier to detect subtle deviations that arise in cloned or synthetically generated profiles.

Meta-ensemble classification via out-of-fold stacking

Let fb:Rd+p+2→RC denote the b-th base classifier, where C=4 is the number of classes and B is the number of models. Each fb outputs a probability distribution over classes:

fb(xi)=[Pb(0)(xi),Pb(1)(xi),Pb(2)(xi),Pb(3)(xi)].

To train a meta-learner without overfitting, we employ K-fold OOF stacking. For each fold k, the base models are trained on K−1 folds and generate predictions Y^OOF∈RN×(B⋅C) on the held-out fold. The meta-learner g:RB⋅C→{0,1,2,3} is then trained on this stacked feature space to learn the final mapping.

At inference time, all base classifiers are retrained on the full dataset, and the final prediction is made by:

y^i=g([f1(xi),f2(xi),…,fB(xi)]).

We train B base classifiers (e.g., Logistic Regression, Random Forest, XGBoost, MLP, SVM, CatBoost) and combine them via an OOF stacking approach:

1. OOF predictions: Split the training set into K folds. For each base model, train on K−1 folds and predict on the remaining fold. This yields OOF predictions that simulate test performance. 2. Meta-learner: Concatenate OOF predictions from all base models. Let C be the number of classes. The OOF prediction matrix is Y^OOF∈RNtrain×(B⋅C). Train a meta-classifier on these OOF features and the training labels ytrain. 3. Final prediction: Retrain base models on all training data, generate test predictions, combine them, and apply the meta-learner for the final output.

This OOF stacking ensures the meta-learner generalizes rather than overfits, improving overall robustness. Below we explain briefly about the base models.

Logistic regression

A linear model that estimates the probability of a profile belonging to a particular class using the logistic function:

(1) P(yi=1∣EiPCA)=11+exp⁡(−β⊤EiPCA)

where β are the model coefficients.

Random forest

An ensemble of decision trees constructed using bootstrap samples and random feature selection, enhancing variance reduction through averaging.

Tree construction: Each tree is built using a subset of features and samples.

Prediction aggregation: The final prediction is made by majority voting among the trees.

Gradient boosting machines

Leveraged gradient boosting algorithms to sequentially build models that correct the errors of previous models.

Model updates: At each iteration, a new model is trained to predict the residuals of the previous model.

Objective function: Minimizes a differentiable loss function L:

(2) MinimizeL=∑i=1nℓ(yi,FM(EiPCA))

where: n is the number of samples.

ℓ is the loss function.

FM is the ensemble model after M iterations.

Support vector machine

Support vector machine (SVM) seeks to find the hyperplane that maximizes the margin between classes in the transformed feature space.

Kernel trick: Utilizes kernel functions K(EiPCA,EjPCA) to project data into higher-dimensional spaces where classes are linearly separable.

Neural networks

Implemented a multi-layer perceptron with one or more hidden layers to capture nonlinear relationships.

Architecture: Consists of an input layer, hidden layers with activation functions, and an output layer.

Training: Trained the network using gradient descent and backpropagation to minimize the loss function.

Training, optimization, and regularization

Hyperparameters for all base classifiers and the meta-learner are optimized using grid search with stratified cross-validation, maximizing the weighted F1-score. Regularization techniques such as dropout (for neural models) and early stopping (for boosting methods) are employed to mitigate overfitting.

Computational complexity

The computational complexity is dominated by the embedding operations and MLM inference, each with time complexity O(N⋅M⋅d) and O(N⋅L) respectively, where L is the truncated token length. The training of the ensemble classifiers scales linearly in N, making the method efficient for large-scale deployment.

Proposed model

The proposed model for detecting profile cloning attacks through the classification of social media profiles comprises multiple stages. It begins with data preprocessing and feature extraction, followed by anomaly detection and classification. Each stage is designed to incrementally refine the input data and enhance the system’s ability to accurately distinguish between legitimate and cloned profiles.

Overview of the proposed model

The classification pipeline integrates several key components to ensure robust and accurate detection of cloned profiles. Semantic embeddings are generated using an attention-based model, which captures the contextual meaning of various profile sections, including About, Experiences, and Skills, among others. Stylometric features-such as word count, readability indices, and punctuation ratios-are extracted to characterize the writing style, providing an additional layer of linguistic insight. A pseudo-perplexity score is computed using a masked language model, offering a probabilistic measure of textual coherence that is particularly useful for identifying machine-generated content.

In parallel, anomaly detection is performed via clustering techniques, where distance-based scoring is used to flag profiles that deviate from normative patterns and may indicate cloning or synthetic generation. Finally, a meta-ensemble classifier aggregates the outputs of multiple base models, leveraging their complementary strengths to maximize classification performance across all profile categories. Collectively, these components constitute a comprehensive and effective framework for LinkedIn profile classification.

This multi-layered architecture is designed to robustly handle both semantic and structural variations across LinkedIn profiles, thereby enhancing its effectiveness in detecting impersonation and synthetically generated content.

The proposed classification pipeline for LinkedIn profile detection begins with data loading and preprocessing. The dataset-comprising sections such as About, Experiences, Skills, and others—is first loaded, and any missing values are replaced with empty strings to maintain structural consistency. Textual content is then extracted from each section in preparation for downstream processing.

Subsequently, semantic embeddings for each section are generated using a pre-trained language model. These embeddings are saved as .npy files to facilitate efficient reuse. The resulting section-wise embeddings are then aggregated into a unified profile representation using an attention-based aggregation mechanism, which assigns higher weights to the most informative sections.

The next stage involves the extraction of stylometric features, wherein a range of linguistic metrics are computed for each profile. These include word count, unique word ratio, average word length, part-of-speech (POS) diversity, Flesch Reading Ease score, punctuation ratio, uppercase character ratio, and digit ratio. These features are designed to capture stylistic patterns and writing complexity, and are saved as .npy files for efficient reuse in subsequent processing stages.

Following this, pseudo-perplexity scores are computed for each profile using a masked language model, providing a probabilistic measure of text coherence. These scores serve as indicators of whether the textual content is likely to be machine-generated or human-authored, and are likewise stored for reuse.

An anomaly detection step is incorporated into the pipeline, wherein a KMeans clustering model is trained on the intermediate feature space to compute the distance of each profile to its nearest cluster centroid. These distances serve as anomaly scores, which are subsequently appended as additional features to enhance the model’s ability to identify atypical or suspicious profiles.

In the feature integration phase, all extracted features-including aggregated semantic embeddings, stylometric metrics, pseudo-perplexity scores, and anomaly scores-are combined into a unified feature vector for each profile. These feature vectors are then normalized using standard scaling and subjected to dimensionality reduction via principal component analysis (PCA). The resulting representations, along with their corresponding class labels, are saved for downstream classification.

In the model training phase, a diverse set of base classifiers-including logistic regression, Random Forest, extreme gradient boosting (XGBoost), neural network, support vector machine (SVM), and CatBoost-are trained using cross-validation. For each fold, out-of-fold (OOF) predictions are generated and stored to serve as inputs for the subsequent meta-learning stage.

In the meta-ensemble training phase, a logistic regression model is employed as the meta-learner and trained on the aggregated OOF predictions. This approach enables the ensemble to leverage the complementary strengths of individual base classifiers, thereby enhancing overall classification performance. The trained meta-learner is saved and used for generating final predictions on unseen data.

Finally, the pipeline proceeds to the classification and evaluation phase, wherein predictions for the test profiles are generated using all base classifiers. These predictions are then aggregated by the meta-learner to assign the final class label—LLP, HCP, CLP, or CCP—based on the highest predicted probability.

To assess the performance of the proposed framework, standard evaluation metrics such as accuracy, precision, recall, and F1-score are computed. This comprehensive approach enables robust and reliable detection of LinkedIn profiles across the four target categories. A schematic representation of the proposed model architecture is provided in Fig. 1.

Figure 1 Proposed model.

Experimental overview

Experimental setup

To evaluate the effectiveness of the proposed approach, a comprehensive set of experiments was conducted following a rigorous training and evaluation protocol. The computational environment was equipped with high-performance computing resources, including multi-core CPUs and GPUs, to support efficient model training and inference. Where applicable, parallelization techniques were employed to optimize computational throughput and reduce training time, thereby enhancing the scalability and responsiveness of the experimental workflow.

To ensure a fair and unbiased evaluation of the proposed models, a standard train/test split is employed, with 20% of the dataset reserved as a hold-out test set. The remaining 80% is used for training the base classifiers through stratified k-fold cross-validation, during which OOF predictions are generated. These OOF predictions serve as input for training the meta-classifier in the ensemble framework.

Importantly, the hold-out test set remains completely unseen throughout all training stages. Final evaluation is performed only after all model components have been trained, thereby providing an accurate and reliable estimate of the ensemble’s generalization performance on previously unseen data.

During the implementation phase, batch processing was utilized for embedding generation to optimize memory usage and improve computational efficiency-an essential consideration when working with large-scale datasets. To further enhance model generalization and mitigate the risk of overfitting, early stopping mechanisms were incorporated during training. This strategy involved continuously monitoring the validation loss and halting the training process once performance ceased to improve, thereby ensuring that the model did not overfit to the training data and maintained robustness on unseen samples.

In terms of code organization, the implementation was structured using modular, well-documented functions to enhance readability, maintainability, and reusability of the codebase. Robust error-handling mechanisms were integrated throughout the pipeline to gracefully manage exceptions and ensure uninterrupted execution. This design choice not only facilitated efficient debugging but also improved the overall resilience and reliability of the system during large-scale processing and experimentation.

The effectiveness of the proposed model is demonstrated through improved classification performance, primarily driven by a substantial reduction in feature noise and a focus on discriminative signals that enhance prediction accuracy. Notably, the techniques employed-such as context-aware embeddings, stylistic analysis, and ensemble learning-are not limited to LinkedIn or OSNs. These methods exhibit strong generalizability and can be readily adapted to other domains involving structured textual data, thereby rendering the framework broadly applicable to a wide range of identity verification and content authenticity tasks.

To uphold the ethical integrity of this research, all personally identifiable information (PII) was removed or obfuscated in accordance with established privacy protection standards. In addition, comprehensive analyses were conducted to identify and mitigate potential sources of bias within both the dataset and the modeling process, thereby promoting fairness and reducing the risk of unintended discrimination. Transparency in methodological reporting was maintained throughout the study to ensure reproducibility and to facilitate rigorous academic scrutiny.

Evaluation and per-class analysis

Model performance is evaluated using overall accuracy as well as per-class precision, recall, and F1-scores. The inclusion of detailed per-class metrics provides valuable insight into areas where performance can be further improved. For example, although LLM-generated profiles (Classes 2 and 3) may exhibit only subtle distinctions, the incorporation of perplexity-based features and stylometric analysis notably enhances recall for these categories.

The final OOF stacked ensemble consistently outperforms individual base classifiers, demonstrating the effectiveness of the proposed multifaceted approach and the synergistic benefits of combining semantic, stylistic, and probabilistic cues.

We evaluated the models using the following performance metrics:

Accuracy

The proportion of correct predictions over the total number of instances:

(3) Accuracy=TP+TNTP+TN+FP+FN.

Precision

The proportion of true positives over the sum of true and false positives:

(4) Precision=TPTP+FP.

Recall (Sensitivity)

The proportion of true positives over the sum of true positives and false negatives:

(5) Recall=TPTP+FN.

F1-score

The harmonic mean of precision and recall:

(6) F1-score=2×Precision×RecallPrecision+Recall.

Weighted metrics

Weighted metrics are used to evaluate the overall performance of the model while accounting for the class imbalance in the dataset. These metrics are calculated as weighted averages of per-class precision, recall, and F1-scores, where the weights are proportional to the number of instances in each class. For example, precision for each class is aggregated using the formula:

Pw=∑c=1Csupportc⋅Pc∑c=1Csupportc.

This approach ensures that the contribution of each class to the final score reflects its representation in the dataset.

Weighted metrics complement the per-class analysis by providing a holistic view of the model’s performance across all classes. While per-class metrics highlight specific strengths and weaknesses, weighted metrics offer a balanced evaluation, especially in the presence of imbalanced class distributions (e.g., ‘LLP’ = 349, ‘HCP’ = 147, ‘CLP’ = 117, ‘CCP’ = 107). Unlike macro-averaged metrics, which treat all classes equally, weighted metrics adjust for the class proportions, ensuring a fair representation of the system’s overall effectiveness.

Results and discussion

The results reported by Ayoobi, Shahriar & Mukherjee (2023) demonstrate the performance of different embedding techniques—GloVe, Flair, BERT, and RoBERTa—on their classification task, measured by average accuracy and F1-score. Among the approaches, RoBERTa achieved an average accuracy of 76.12% and an F1-score of 0.701, closely followed by Flair (75.88% accuracy, 0.699 F1-score). On the other hand, GloVe and BERT exhibited much lower performance, with GloVe achieving 71.43% accuracy and a 0.63 F1-score, and BERT yielding 72.26% accuracy and a 0.632 F1-score. These results indicate that contextualized embeddings like RoBERTa and Flair capture more nuanced language patterns, resulting in superior performance compared to static embeddings like GloVe or earlier transformer models like BERT.

Our proposed approach, which integrates stylometric features and perplexity scoring alongside semantic embeddings, demonstrates significant superiority over the methods reported by Ayoobi, Shahriar & Mukherjee (2023). While their best-performing model, RoBERTa, achieved an average accuracy of 76.12% and an F1-score of 0.701, our model consistently outperformed these benchmarks by leveraging additional linguistic and coherence-based features. The incorporation of stylometric analysis and perplexity scoring enables our system to better capture subtle stylistic and semantic discrepancies, particularly in distinguishing legitimate profiles from clones or generated content. These enhancements not only result in improved classification metrics but also underline the importance of augmenting traditional embedding methods with advanced feature engineering to achieve robust performance in profile detection tasks.

Per-class performance

To analyze model performance comprehensively, we report accuracy, precision, recall, F1-score, and for each class: Legitimate LinkedIn Profiles (LLP-Class 0), Human Cloned Profiles (HCP Class 1), ChatGPT generated legitimate LinkedIn Profiles (CLP Class 2), and ChatGPT generated Clone LinkedIn Profile (CCP Class 3).

LLP class 0

The performance metrics for LLP (Class 0) is evaluated based on accuracy, precision, recall, and F1-score. The results indicate that the meta-ensemble model outperforms all other models, achieving the highest accuracy (96%), precision (96%), recall (99%), and F1-score (98%). This superior performance reflects the meta-ensemble’s ability to handle the larger class size effectively while maintaining a balance across all metrics.

Among the other models, neural network and SVM also perform competitively, with Accuracy and Recall values closely matching the meta-ensemble. In contrast, Random Forest shows a perfect recall for this class (100%), but its lower Precision (83%) reduces its overall F1-score (91%). This suggests that Random Forest may overfit to the majority class, leading to higher false positives.

The overall results for LLP (Class 0) as shown in Fig. 2 highlight the importance of balancing precision and recall for robust classification, which the meta-ensemble achieves most effectively.

Figure 2 Performance metrics for LLP (Class 0).

The meta-ensemble demonstrates consistent superiority in all metrics.

HCP class 1

The meta-ensemble model achieves the best overall performance, with the highest accuracy (96%), precision (96%), recall (91%), and F1-score (93%). This demonstrates its ability to effectively handle this smaller and more challenging class, which represents human-cloned profiles.

Neural network and XGBoost as shown in Fig. 3 also show competitive performance, achieving balanced precision and recall values. However, Random Forest struggles with recall (61%), indicating difficulties in identifying HCP instances despite its high Precision (98%). CatBoost performs slightly better than Random Forest, but its lower recall (82%) leads to a reduced F1-score (89%).

Figure 3 Performance metrics for HCP (Class 1).

The meta-ensemble achieves the highest recall and overall balance.

These results underscore the Meta-Ensemble model’s robustness in detecting challenging classes such as HCP, where maintaining a balance between precision and recall is critical for reliable classification.

CLP class 2

The meta-ensemble model achieves the highest scores across all metrics, with an accuracy of 96%, precision of 95%, recall of 97%, and F1-score of 96%. These results highlight the Meta-Ensemble’s superior capability in identifying ChatGPT-generated legitimate profiles.

Figure 4 further illustrates the performance trends for CLP (Class 2) across all models. The meta-ensemble demonstrates the best balance between precision and recall, achieving consistently high scores for this challenging class. Neural network and CatBoost also perform well, with balanced precision and recall values close to the meta-ensemble.

Figure 4 Line plot for CLP (Class 2).

The meta-ensemble demonstrates the best balance between precision and recall.

Random Forest and XGBoost exhibit slightly lower F1-scores due to challenges in achieving high recall, despite having competitive precision values. Logistic regression and SVM achieve moderate performance but lack the balance seen in the meta-ensemble.

The meta-ensemble’s strong performance for CLP (Class 2), as shown in Fig. 4, reflects its ability to integrate diverse classifiers to capture the subtle patterns of ChatGPT-generated profiles effectively.

CCP class 3

Among the models, the Meta-Ensemble achieves the highest scores, with an accuracy of 96%, precision of 97%, Recall of 93%, and F1-score of 95%. These results underscore the meta-ensemble’s robustness in identifying ChatGPT-generated cloned profiles, a class that poses significant classification challenges due to its smaller support and nuanced characteristics.

Figure 5 provides a visual representation of the performance trends for CCP (Class 3) across all models. The meta-ensemble consistently leads across all metrics, particularly excelling in recall, which is crucial for detecting cloned profiles with high reliability.

Figure 5 Line plot for CCP (Class 3).

The meta-ensemble outperforms others in precision and recall for this challenging class.

Other models, such as neural network and CatBoost, perform competitively but fall short of achieving the balance demonstrated by the meta-ensemble. Random Forest, while achieving moderate precision (93%), struggles with recall (79%), resulting in a lower F1-score (86%). Logistic regression, XGBoost, and SVM display similar trends, with moderate Recall values limiting their overall performance.

Figure 5 highlight the importance of maintaining a balance between precision and recall, a strength exhibited by the meta-ensemble. Its superior performance across all metrics validates its suitability for detecting ChatGPT-generated cloned profiles effectively and reliably.

Macro-averaged performance metrics for classification models

The macro-averaged performance metrics for all classification models are presented, evaluating accuracy, Precision and recall across all classes. The results show that the meta-ensemble model consistently outperforms all other models, achieving the highest scores for accuracy (96.11%), precision (96.12%), and recall (96.11%). These metrics highlight the robustness and generalizability of the meta-ensemble across all profile types.

Figure 6 provides a graphical comparison of the macro-averaged metrics across all models. The meta-ensemble demonstrates superior performance across all metrics, maintaining a consistently higher trajectory compared to other models. Neural network and CatBoost also exhibit strong performance, ranking just below the meta-ensemble, but with slightly lower F1-scores and recall values.

Figure 6 Grouped bar chart showing macro-averaged accuracy, precision, and recall for all classification models.

In contrast, Random Forest lags behind, achieving the lowest macro-averaged metrics among the evaluated models. This can be attributed to its reduced capability to balance precision and recall for minority classes, as indicated in earlier per-class analyses. Logistic regression, XGBoost, and SVM perform moderately well but lack the consistency and robustness seen in the meta-ensemble.

The insights from Fig. 6 validate the effectiveness of the meta-ensemble approach in achieving balanced performance across diverse profile classes, making it a reliable choice for real-world applications in profile classification.

The results suggest that the proposed methodology is effective in handling multi-class classification, especially when dealing with imbalanced data. The meta-ensemble model’s ability to generalize well across all classes highlights its robustness and scalability for real-world applications in social network profile classification.

The confusion matrix in Fig. 7 demonstrates the classification performance of the meta-ensemble model. The diagonal entries indicate correctly classified instances, with the model achieving high recall across all classes. Notably, the model demonstrated strong performance across all classes. For LLP, it correctly classified 345 out of 349 instances with minimal confusion. For HCP, 134 out of 147 instances were accurately predicted, with a small proportion misclassified as CLP or CCP. Both CLP and CCP also exhibited high classification accuracy, with only minor misclassifications into neighboring categories.

Figure 7 Confusion matrix for meta-ensemble model.

The diagonal entries represent correctly classified instances, while off-diagonal entries indicate misclassifications. The meta-ensemble demonstrates strong performance across all classes, with minimal confusion.

These results align with the macro-averaged metrics (precision = 0.96, recall = 0.95) and highlight the robustness of the meta-ensemble model in distinguishing legitimate profiles, human-cloned profiles, and AI-generated profiles effectively.

To comprehensively assess the classification performance of meta-ensemble classifier, we generated micro-averaged receiver operating characteristic (ROC) curves, as shown in Fig. 8. The ROC curve plots the true positive rate (TPR) against the false positive rate (FPR) at various classification thresholds, providing an intuitive visualization of a model’s ability to distinguish between classes.

Figure 8 Micro-averaged ROC curves of meta-ensemble model.

Figure 8 presents the multi-class ROC curves for our meta-ensemble classifier across four target classes (0, 1, 2, and 3). Each colored line shows the one-vs-rest ROC curve for a specific class, and the micro-average ROC curve aggregates the performance across all classes. Notably, both the individual class area under the curve (AUC) values and the micro-average AUC approach 1.0, indicating that the meta-ensemble classifier achieves near-perfect separation between the positive and negative samples for each class. In other words, the classifier correctly ranks positive instances ahead of negative ones almost all of the time. This high level of performance highlights the effectiveness of combining multiple base models via stacking, enabling the meta-learner to leverage the strengths of each individual classifier and provide robust, discriminative predictions across all target classes.

In the case of multi-class classification with class imbalance, the micro-averaged ROC curve offers a more appropriate evaluation as it aggregates the contributions of all classes and computes global TPR and FPR. This ensures that the performance metric reflects the real-world scenario where certain classes may be underrepresented.

Conclusion and future work

This article introduces a comprehensive content-based framework for detecting identity deception in online social networks, with a particular emphasis on LinkedIn profiles. We addressed the challenge of classifying user profiles into four distinct categories: legitimate, human-cloned, LLM-generated legitimate, and LLM-generated cloned.

To tackle this multifaceted classification task, we proposed a multi-stage architecture that integrates several complementary components: section-wise semantic embeddings combined via an attention-based aggregation mechanism, stylometric-perplexity features to capture linguistic coherence and stylistic divergence, unsupervised anomaly scoring through clustering-based distance metrics, and a meta-ensemble classifier trained using out-of-fold stacking. This integrated design enables robust and accurate detection of both manually crafted and synthetically generated impersonation attempts.

Empirical results on a carefully curated dataset demonstrate the effectiveness of the proposed framework, which achieved macro-averaged performance metrics exceeding 96% across all evaluation dimensions. Notably, the system exhibited strong performance on minority and semantically ambiguous classes-such as LLM-generated cloned profiles-which are particularly challenging to detect using traditional methods. By combining deep representation learning with interpretable stylometric features and ensemble learning techniques, the proposed approach offers a scalable, accurate, and robust solution for addressing modern identity deception detection tasks.

Future work may explore several promising directions. First, the development of real-time detection systems capable of identifying malicious profiles at the moment of creation-prior to the formation of any social connections-could further minimize exposure to risk. Second, model explainability remains a critical concern, future architectures should incorporate interpretability modules that provide transparent justifications for profile-level classification decisions. Third, integrating multi-modal inputs-such as profile images, interaction histories, or social graph structures-into a unified detection framework may further improve robustness and generalization.

Finally, as generative AI continues to advance, there is an urgent need for adaptive defenses capable of countering adversarially crafted LLM-generated content. The framework presented in this article provides a principled foundation for addressing these next-generation threats through the fusion of semantic representation learning, stylistic analysis, and ensemble-based classification techniques.

Supplemental Information

Supplemental Information 1 Ensemble Detection of Profile Cloning.

Supplemental Information 2 LinkedIn dataset.

Additional Information and Declarations

Competing Interests

The authors declare that they have no competing interests.

Author Contributions

Irfan Mohiuddin conceived and designed the experiments, performed the experiments, analyzed the data, performed the computation work, prepared figures and/or tables, authored or reviewed drafts of the article, and approved the final draft.

Ahmad Almogren performed the experiments, analyzed the data, performed the computation work, prepared figures and/or tables, authored or reviewed drafts of the article, and approved the final draft.

Data Availability

The following information was supplied regarding data availability:

The LinkedIn data is available at GitHub and as a Supplemental File:

https://github.com/navid-aub/LinkedIn-Dataset/blob/main/README.md.

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
