# Peer review of "Ensemble techniques for detecting profile cloning attacks in online social networks"

_PeerJ Computer Science, doi:10.7717/peerj-cs.3182_

## Round 0.1 · original submission · Major Revisions

This work is valuable but also has some weaknesses. Please consider the reviewer's comments carefully and revise the article accordingly.

Reviewer 1 ·

Basic reporting

Authors presented ensemble method for profile cloning detection.

Experimental design

very poor, see full comments.

Validity of the findings

could be improved.

Additional comments

Authors presented an ensemble method for profile cloning detection, however, I have the following observations:
In section 3, Data Collection, the authors mentioned collecting data from social network sites and annotating each profile with a target variable that specifies the category, such as legitimate profiles, fake profiles, and LLM-generated real/fake profiles. However, they did not mention the mechanism deployed by them. Did they label profiles manually, or was some automated mechanism used?
-The size of the data set (3600 profiles) is too small to achieve a generalized model. There’s no mention of the Train/Test split of the data.
The first block of Figure 1 shows various Social Network icons such as LinkedIn, Facebook, etc. Did they mix profiles from various platforms? LinkedIn is a professional networking site, while Facebook is a personal experience sharing platform. This further dilutes the point of generalizability of models.
-To show performance of model, it’s better to stick to RoC plots/ Precision-recall plots etc.
-The context-aware features used to represent profiles must be visualized using tSNE.

Reviewer 2 ·

Basic reporting

This paper is valuable in the research area for detecting profile cloning attacks in online social networks. However, some parts undervalue the manuscript in several aspects.
1. The abstract is shallow. It can be described clearly and sharply. The author should explain in the abstract the main problem of the current method in IDS and why this study is essential. What is the final result and its contribution in a short description?
2. In the introduction part, description for the novelty and significance of the study is too shallow. The author should describe them in more detail.
3. Background study is poor. The author should improve the related works part with more updated and fit documents
4. This paper merely implements the standard machine learning algorithms to construct a detection model. Moreover, I can’t find the significance of this study, and the manuscript lacks novelty in the proposed technique. If the architecture is novel, the authors should describe the novelty in the introduction as the originality of the proposed techniques.

Experimental design

5. The author should explain the math formulation of the proposed technique function in depth. Besides, what are the influences of the proposed architecture on enhancing the detection process?

Validity of the findings

6. Authors must conduct more ablation studies to compare the detection performance of the proposed architecture to other standard machine learning methods in the same environment.
7. English presentation is poor and must be improved in many areas.
8. The conclusion is tiny and shallow, and it should be greatly improved. What is the next trend in this matter? The author must show it.
9. References are tiny. The author must update with more current and updated articles to support the manuscript.

---

## Round 0.2 · Minor Revisions

Thanks for the effort to improve the work. Some issues have been addressed. However, the authors should explain several contributions of the proposed method clearly and detailedly, and also improve the English writing.

**Language Note:** The review process has identified that the English language must be improved. PeerJ can provide language editing services - please contact us at [email protected] for pricing (be sure to provide your manuscript number and title). Alternatively, you should make your own arrangements to improve the language quality and provide details in your response letter. – PeerJ Staff

Reviewer 2 ·

Basic reporting

This manuscript addresses the drawbacks according to the reviewer's comments. However, athor should explain several contribution of the proposed method in clear and detail.

Experimental design

The author has improved experimental design of this manuscript.

Validity of the findings

no comment

Additional comments

This manuscript should improve its English presentation.

---

## Round 0.3 · accepted · Accept

Thanks to the authors for their efforts to improve the work. I believe this version has been revised according to the comments. Thus I recommend to accept it.